# Phosphorylation and activation of ubiquitin-specific protease-14 by Akt regulates the ubiquitin-proteasome system

**Daichao Xu[1,2], Bing Shan[1†], Byung-Hoon Lee[3†], Kezhou Zhu[1†], Tao Zhang[1†], Huawang Sun[1], Min Liu[1], Linyu Shi[1], Wei Liang[1], Lihui Qian[1], Juan Xiao[1‡], Lili Wang[1], Lifeng Pan[2], Daniel Finley[3], Junying Yuan[1,3]***

[1]Interdisciplinary Research Center on Biology and Chemistry, Shanghai Institute of Organic Chemistry, Chinese Academy of Sciences, Shanghai, China; [2]State Key Laboratory of Bioorganic and Natural Products Chemistry, Shanghai Institute of Organic Chemistry, Chinese Academy of Sciences, Shanghai, China; [3]Department of Cell Biology, Harvard Medical School, Boston, United States

**Abstract** Regulation of ubiquitin-proteasome system (UPS), which controls the turnover of short-lived proteins in eukaryotic cells, is critical in maintaining cellular proteostasis. Here we show that USP14, a major deubiquitinating enzyme that regulates the UPS, is a substrate of Akt, a serine/threonine-specific protein kinase critical in mediating intracellular signaling transducer for growth factors. We report that Akt-mediated phosphorylation of USP14 at Ser432, which normally blocks its catalytic site in the inactive conformation, activates its deubiquitinating activity in vitro and in cells. We also demonstrate that phosphorylation of USP14 is critical for Akt to regulate proteasome activity and consequently global protein degradation. Since Akt can be activated by a wide range of growth factors and is under negative control by phosphoinosotide phosphatase PTEN, we suggest that regulation of UPS by Akt-mediated phosphorylation of USP14 may provide a common mechanism for growth factors to control global proteostasis and for promoting tumorigenesis in PTEN-negative cancer cells.

*For correspondence: jyuan@hms.harvard.edu

†These authors contributed equally to this work

Present address: ‡Youjiang Medical University for Nationalities, Guangxi, China

Competing interests: The authors declare that no competing interests exist.

## Introduction

The ubiquitin–proteasome system (UPS), a major degradative mechanism in eukaryotic cells, is involved in the degradation of short-lived proteins as well as misfolded and damaged proteins (*Komander and Rape, 2012*). The 26S proteasome specifically targets and degrades proteins conjugated to ubiquitin. Regulation of protein deubiquitination by deubiquitinating enzymes (DUBs) is recognized as an important regulatory step in the UPS. Ubiquitin-specific protease-14 (USP14), a DUB reversibly associated with the proteasome, negatively regulates the activity of proteasomes by trimming ubiquitin chains on proteasome-bound substrates (*Borodovsky et al., 2001; Koulich et al., 2008; Lee et al., 2010*). Purified recombinant USP14 is largely inactive and can be highly activated when in association with proteasome (*Hu et al., 2005; Koulich et al., 2008; Lee et al., 2010*). However, a significant fraction of USP14 is present intracellularly in a proteasome-free state (*Koulich et al., 2008*), and it is not clear if and how proteasome-free USP14 might serve a significant physiological function.

Akt, a serine/threonine-specific protein kinase and an important intracellular signaling transducer for growth factors such as insulin, is involved in regulating cell proliferation, metabolism,

**eLife digest** Proteins are the workhorses of cells. These molecules provide structure, transmit messages and carry out many other essential tasks. When proteins have fulfilled their purpose, or become damaged, they must be removed through a garbage disposal-like molecular machine in cells called the proteasome. A breakdown in the proteasome may lead to diseases in humans such as cancers and neurodegeneration. Cells have a system that can identify and mark proteins for destruction, and another system that counteracts this process and spares proteins from destruction. Precise regulation of these two systems helps ensure a healthy balance in cells.

One enzyme that can spare proteins from the proteasome is called USP14. Previously, this enzyme is known to be switched on when it connects with the protein disposal machinery to control which proteins get destroyed. But, many of the USP14 enzymes in cells are not associated with this proteasome machinery and it was unclear if and how these 'free' enzymes might be important for the cell.

Now, Xu et al. report a new mechanism that can switch on USP14: another enzyme called Akt can switch on USP14 by adding a phosphate group to a specific site in USP14. Akt is an important signaling molecule that is activated in many tumor cells to promote the growth and multiplication of cells. Xu et al. discovered that by controlling USP14 activity, Akt can control the activity of the protein disposal machinery that in turn regulates the levels of many other proteins. These findings suggest that abnormal activity of USP14 in tumor cells with elevated Akt activity may contribute to cancer formation.

transcription, migration, and apoptosis (*Manning and Cantley, 2007*). The activity of Akt is regulated by PI(3,4,5)P3, a lipid product of the phosphoinositide 3-kinases (PI3Ks). The intracellular levels of PI(3,4,5)P3 are negatively regulated by phosphatases such as SHIP1/2 and phosphatase and tensin homolog (PTEN). The latter, a phosphoinoside phosphatase, is encoded by a tumor suppressor gene that is mutated in human cancers at high frequency (*Cantley and Neel, 1999*). Akt has been reported to mediate the phosphorylation of many substrates that in turn regulate cell proliferation, metabolism, transcription, migration, and apoptosis. However, very little is known about its role in the UPS, and furthermore no mechanistic link between Akt and UPS has been elucidated.

In this study, we report that USP14 is an Akt substrate and that this phosphorylation activates the DUB activity of USP14 both in vitro and in cells. We also demonstrate that phosphorylation of USP14 is critical for Akt to control UPS and consequentially global protein degradation via the UPS. Our study reveals a novel mechanistic connection between activated Akt and the UPS through regulating of USP14 phosphorylation, which can impact global proteostasis in PTEN-negative cancer cells.

## Results

### Phosphorylation of USP14 on Ser432 by Akt

Two forms of USP14 have been determined crystallographically: the inactive free form and an adduct between Ub-aldehyde (Ubal) and USP14, which provides insight into the catalytically active state (*Hu et al., 2005*). The key difference between these two structures is in the position of the blocking loops, BL1 and BL2, which project over the catalytic cleft of USP14 and block the access of the C-terminal residues of ubiquitin in the inactive form (*Figure 1A*). In Ubal-modified USP14, BL1 and BL2 are rearranged, thus exposing the cleft. In particular, Ser432, located within BL2, shifts its position over a distance of 3–5 Å between the two states (*Hu et al., 2005*) (*Figure 1B*).

Since Ser432 residue is located very close to a highly negatively charged patch (*Figure 1C*), we reasoned that when Ser432 residue was phosphorylated, the negatively charged phosphate group might induce a repulsive force, thereby inducing rearrangement of the BL2 loop and removing the inhibitory effect of this loop on the activity of USP14. The amino acid sequences around Ser432 are highly evolutionarily conserved among USP14 orthologs (*Figure 1D*) and Ser432 is predicted to be an Akt substrate by Scansite (http://scansite3.mit.edu/#home). We therefore tested the possibility that USP14 might be a substrate of activated Akt. We first examined the interaction between USP14

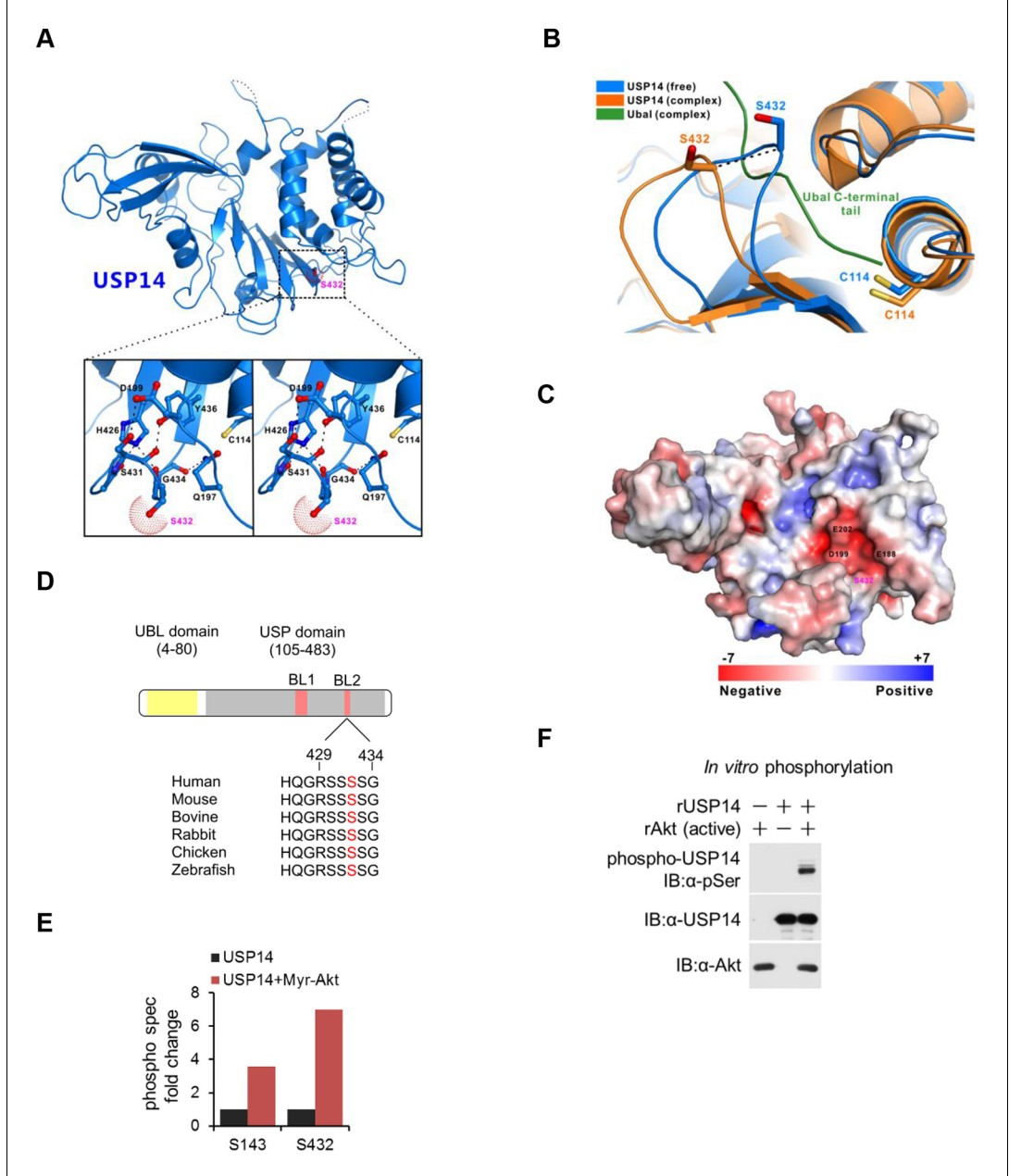

**Figure 1.** Structural basis of ubiquitin-specific protease-14 (USP14) activation by phosphorylation of Ser432. (**A**) Detailed view of blocking loop 2 (BL2), which occludes the active site of USP14 (PDB access code 2AYN). The BL2 loop, which contains Ser432, is shown in stick model, in the apo form. (**B**) Combined ribbon representation and stick model showing a comparison of the conformations of the BL2 loop contained in the apo form (blue, PDB access code 2AYN) and in the USP14- Ub-aldehyde (Ubal) adduct (orange, PDB access code 2AYO). In this drawing, the Ser432 and Cys114 residues are shown in stick model, and the bound Ubal (a ubiquitin derivative in which the C-terminal carboxylate is replaced by an aldehyde) in the complex is drawn in green. (**C**) A surface charge potential representation (contoured at ± 7 kT/eV; blue/red) of USP14 (PDB accession 2AYN) showing that the S432 residue is very close to a highly negatively charged patch mainly formed by the acidic E188, D199, and E202 residues. When S432 is phosphorylated, the negatively charged phosphate group may induce a repulsive force, thereby relieving inhibition of the catalytic activity of USP14. (**D**) USP14 domain organization and sequence alignment of the Akt phosphorylation site within USP14 orthologs from different species. Two BLs (BL1 and BL2) covering the USP14 active site are shown. The Akt phosphorylation site in USP14 from different species as predicted by Scansite. (**E**) S432 is the major phosphorylation site in USP14. HEK293T cells were treated as in *Figure 1—figure supplement 1B*, followed by electrospray ionization mass spectrometry (ESI-MS) analysis. Spectral counts were determined by ESI-MS. (**F**) Akt phosphorylates USP14 in vitro. Bacterially expressed and purified USP14 was incubated with active Akt in the presence of ATP. Reaction products were resolved by sodium dodecyl sulfate polyacrylamide gel electrophoresis (SDS-PAGE), and phosphorylated species were detected by a phospho-Ser antibody.

*Figure 1 continued on next page*

*Figure 1 continued*

The following figure supplement is available for figure 1:

**Figure supplement 1.** Akt phosphorylates ubiquitin-specific protease-14 (USP14).

and Akt using a co-immunoprecipitation assay. As shown in *Figure 1—figure supplement 1A*, when USP14 and Akt were overexpressed in HEK293T cells, their interaction was readily detectable. To test whether Akt could phosphorylate USP14, we overexpressed USP14 and an activated Akt (Myr-Akt) in HEK293T cells, and performed a quantitative phosphoproteomic analysis (*Figure 1—figure supplement 1B*). We identified four phosphorylation sites on USP14 when it was expressed alone: Ser143, Ser230, Thr235, and Ser432 (*Figure 1—figure supplement 1C,D*). Notably, the phosphorylation levels of two of the four sites, Ser143 and Ser432, were increased considerably in cells expressing activated Akt (*Figure 1E*).

To examine whether USP14 is a direct substrate for Akt, we conducted an in vitro kinase assay using activated recombinant Akt and purified recombinant USP14 expressed in *Escherichia coli*. We found that co-incubation of USP14 and Akt led to modification of USP14 as detected by a pan phospho-Ser antibody (*Figure 1F*), suggesting that USP14 is a substrate for Akt.

To determine if Ser143 and Ser432 were indeed phosphorylated by Akt, we used this pan phospho-Ser antibody as above and found phosphorylation of wild type (WT) USP14, but not of S143A/S432A mutant USP14, after incubating with activated Akt in a kinase assay (*Figure 2A*). To differentiate the relative importance of Ser143 and Ser432 as phosphorylation sites by Akt, we overexpressed activated Akt (Myr-Akt) in HEK293T cells with WT, S143A, S432A, or double S143A/S432A (AA) mutants. We found that S143A mutant showed partially reduced phosphorylation as compared to that of WT, whereas phosphorylation of the USP14 S432A mutant was significantly decreased and that of AA double mutant was completely eliminated (*Figure 2B*). These results suggested S432 as a major and S143 as a minor phosphorylation site of Akt.

The phosphorylation of USP14 by Akt was further confirmed using an Akt phosphorylation-consensus motif (R××S/T) antibody (*Figure 2—figure supplement 1A*). The reactivity of USP14 with pan phospho-Ser antibody was eliminated after incubation with lambda phosphatase (*Figure 2C*). Notably, the phosphorylation levels of USP14 were decreased in cells when treated with MK2206, an inhibitor of Akt (*Figure 2D*), or when serum deprived, a condition known to inactivate endogenous Akt (*Zhang et al., 2015*) (*Figure 2D*).

To further verify the phosphorylation of USP14 S432 by Akt, we developed a phospho-Ser432-specific antibody. Phosphorylation of S432 can be detected after incubation of WT, but not S432A mutant USP14, with recombinant activated Akt in a kinase reaction (*Figure 2E*). This was further confirmed by using phos-tag electrophoresis which can specifically retard the migration of phosphorylated protein species (*Kinoshita et al., 2009*) (*Figure 2E*). Expression of Myr-Akt also led to S432 phosphorylation of endogenous USP14 (*Figure 2F*). Treatment with either MK2206 or AZD5363, two structurally unrelated Akt inhibitors, led to decrease of USP14 S432 phosphorylation levels (*Figure 2—figure supplement 1B,C*). Moreover, treatment with PI3K inhibitors, either Wortmannin or GDC0941, but not ERK1/2 inhibitor U0126, also significantly decreased the phosphorylation levels of USP14 S432 (*Figure 2—figure supplement 1D,E*). In addition, we tested growth factors such as insulin-like growth factor (IGF-1) or epidermal growth factor (EGF), both of which are known to promote activation of Akt. We found that the treatment of IGF-1 or EGF resulted in phosphorylation of USP14 S432, which was blocked in cells pretreated with MK2206 (*Figure 2G,H*). Finally, USP14 S432 is dramatically more phosphorylated in PTEN knockout mouse embryonic fibroblasts (MEFs), which carry high levels of Akt activity, than that of WT MEFs as determined by western blotting using the phospho-USP14(S432) antibody and phos-tag electrophoresis (*Figure 2I*), and the phosphorylation of USP14 S432 was blocked by Akt inhibitors (*Figure 2—figure supplement 1F*). From these results, we conclude that Ser432 of USP14 is a major phosphorylation site by Akt.

## Activation of USP14 by Akt-mediated phosphorylation

Because bacterially expressed and purified USP14 protein exhibits very low catalytic activity (*Lee et al., 2010*), we tested whether Akt-mediated phosphorylation might activate the DUB activity of USP14. We compared the activity of recombinant USP14 in a Ub-AMC (ubiquitin-7-amido-4-

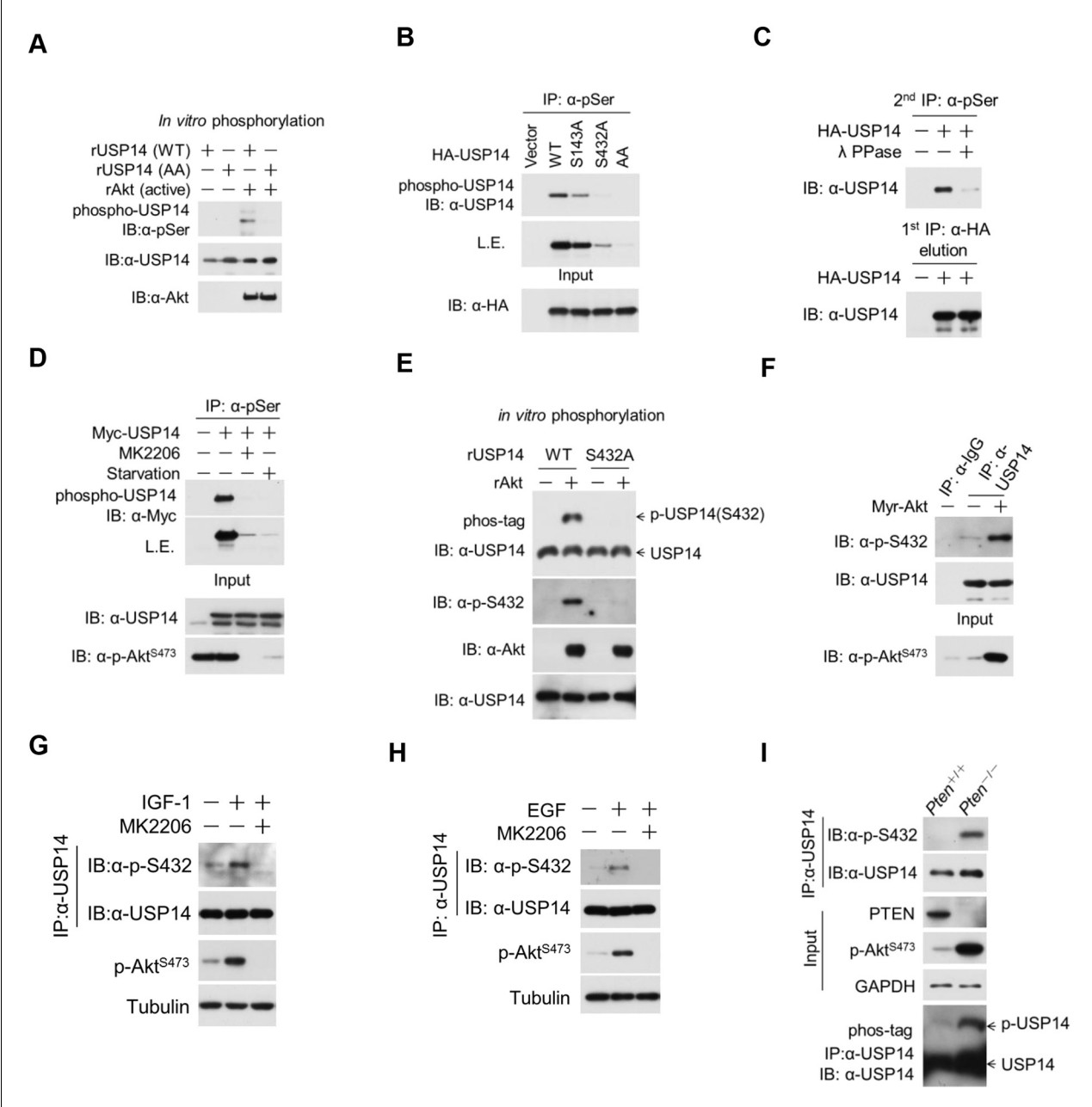

**Figure 2.** Ubiquitin-specific protease-14 (USP14) is phosphorylated at Ser432 by activated Akt. (**A**) In vitro phosphorylation of USP14 at S432 by Akt. Bacterially expressed and purified wild type USP14 or AA mutant incubated with active Akt in the presence of ATP. Reaction products were resolved by sodium dodecyl sulfate polyacrylamide gel electrophoresis (SDS-PAGE), and phosphorylation was detected by the phospho-Ser antibody. (**B**) Akt phosphorylates USP14 at S432 in vivo. Western blot analysis of whole cell lysate and immunoprecipitates derived from HEK293T cells transfected with wild type USP14, USP14 S143A, USP14 S432A, and USP14 S143A/S432A (AA) constructs using the phospho-Ser antibody. L.E., long exposure. (**C**) Immunoprecipitation (IP) and IB analysis of HEK293T cells transfected with HA-USP14 and Myr-Akt and preincubated with or without λ-phosphatase as indicated. (**D**) Inhibition of Akt decreased exogenous USP14 phosphorylation. HEK293T cells were transfected with Myc-USP14 for 20 hr then treated with 1 μM MK2206 or deprived of serum for another 4 hr before harvest. (**E**) In vitro kinase assay to detect Akt phosphorylation of USP14 by phospho-Ser432-specific antibody and phos-tag-containing gels. Bacterially expressed and purified wild type USP14 or S432A mutant was incubated with active Akt in the presence of ATP. The reaction products were resolved by SDS-PAGE, and USP14 phosphorylation was detected using an antibody that specifically recognizes Ser432 phosphorylation of USP14 or determined by differential migration on phos-tag gels. (**F**) In vivo detection of endogenous USP14 Ser432 phosphorylation by anti-p-Ser432-specific antibody. Western blot analysis of immunoprecipitates derived from H4 cells transfected with or without Myr-Akt plasmids using the anti-p-Ser432-specific antibody. (**G, H**) Phosphorylation of endogenous USP14 S432 upon stimulation with insulin-like growth factor (IGF-1) or epidermal growth factor (EGF). HEK293T cells were serum-starved and pretreated with Akt inhibitor MK2206 (1 μM) for 30 min before stimulation with IGF-1 (100 ng/mL) for 30 min (**G**) or EGF (100 ng/mL) for 1 hr (**H**). The cell lysates were immunoprecipitated with USP14

*Figure 2 continued on next page*

*Figure 2 continued*

antibody and western-blotted with anti-p-S432 antibody. (I) Phosphorylation of endogenous USP14 S432 in *Pten* knockout cells with high activity of Akt. Lysates from mouse embryonic fibroblasts (MEFs) with indicated genotypes were immunoprecipitated with USP14 antibody and then Western blotted with p-S432 antibody. The differential migration of phospho-USP14 on phos-tag-containing gels was determined as shown in the bottom panel.

The following figure supplement is available for figure 2:

**Figure supplement 1.** Ubiquitin-specific protease-14 (USP14) is phosphorylated at Ser432 by Akt.

methylcoumarin, a fluorogenic substrate) hydrolysis assay in the presence or absence of Akt. Bacterially expressed and purified USP14 (*Figure 3—figure supplement 1*) showed trace hydrolyzing activity towards Ub-AMC as reported (*Lee et al., 2010*), while USP14 incubated with Akt showed high activity (*Figure 3A*). To validate Akt-mediated activation of USP14 in cells, we coexpressed USP14 and Myr-Akt in HEK293T cells. USP14 immunoprecipitated from cells coexpressing activated Akt showed higher activity in Ub-AMC assay than that expressed alone (*Figure 3B*). On the other hand, USP14 isolated from HEK293T cells incubated with Akt inhibitor MK2206 showed reduced activity in Ub-AMC assay (*Figure 3C*). Moreover, USP14 isolated from HEK293T cells stimulated with IGF-1 showed higher activity, which was suppressed when cells were pretreated with MK2206 (*Figure 3D*). To determine the specific contribution of Ser432, we compared the activity of USP14 S432A mutant protein in Ub-AMC assay with that of WT in the presence of Akt, and found that the stimulating effect of Akt on the hydrolyzing activity of USP14 was largely blocked by S432A mutation (*Figure 3E*), but not by S143A mutation (*Figure 3—figure supplement 2B*).

To further characterize the effect of Ser432 phosphorylation, we expressed and purified recombinant S432E USP14 protein, which mimics the phosphorylation state of USP14, from *E. coli* (*Figure 3—figure supplement 1*) and analyzed its activity by Ub-AMC assay. Interestingly, we found that USP14 S432E mutant protein alone showed high levels of Ub-AMC hydrolyzing activity (*Figure 3F*). Consistent with S432 as the major phosphorylation site by Akt, double E mutant (S143E/S432E) showed almost the same levels of hydrolyzing activity as that of S432E single mutant and S143E mutation had no significant impact on the activity of USP14 (*Figure 3—figure supplement 2C,D*). To determine its enzyme kinetics, we incubated USP14 S432E mutant protein with increasing amounts of Ub-AMC (*Figure 3—figure supplement 2E*) and determined the $K_m$ value ($K_m = 26$ μM) from the slope of a Lineweaver–Burk plot (*Figure 3G*).

We characterized the distributions of p-S432 USP14 and total USP14 with that of proteasome in $Pten^{-/-}$ MEFs using glycerol gradient centrifugation (*Koulich et al., 2008*). We found that the majority of p-S432 USP14 was distributed in the fractions with lower molecular weight proteins and distinguishable from the fractions where larger protein complexes, such as proteasomes, were localized. On the other hand, unphosphorylated USP14 was found in the fractions where larger molecular weight complexes, such as proteasome, are known to be localized (*Figure 3—figure supplement 2F*). Thus, S432 phosphorylated and unphosphorylated USP14 might be distributed differently in the cells. We next determined whether phospho-mimetic mutant of USP14 could be further activated by interacting with proteasome. Interestingly, we found that the Ub-AMC hydrolytic activity of S432E mutant could be further activated when incubated with proteasome in vitro (*Figure 3H*). Taken together, these results suggest that S432 phosphorylation and interaction with proteasome may be two different regulatory mechanisms for USP14.

## Phosphorylation of USP14 promotes both K48 and K63 deubiquitination activity

To assess the impact of USP14 phosphorylation on its selectivity towards different types of ubiquitin linkages, we incubated USP14 WT and S432E mutant protein with diubiquitin species of K48, K63, and linear linkages. Conversion to monomeric Ub was monitored via sodium dodecyl sulfate polyacrylamide gel electrophoresis (SDS-PAGE) followed by western blotting. We observed significantly increased hydrolytic activity of S432E mutant, as compared to that of WT, towards both Lys48 and Lys63 diubiquitin, while linear diubiquitin was not readily cleaved by WT or mutant USP14 (*Figure 4A,B* and *Figure 4—figure supplement 1A*). Similarly, immunoprecipitated USP14 from cells showed significant activity toward both Lys48 and Lys63 diubiquitin, but not linear diubiquitin

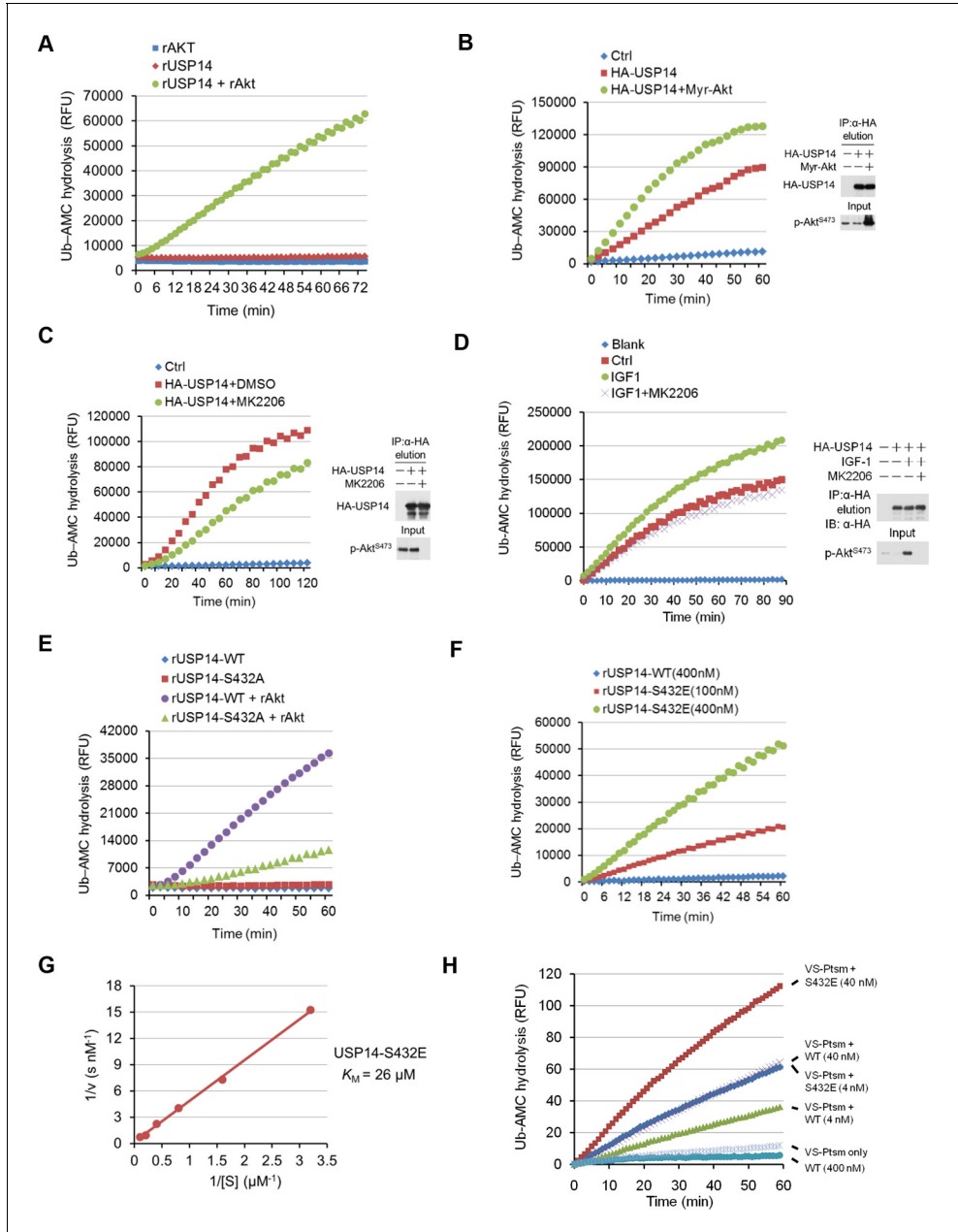

**Figure 3.** Phosphorylation of ubiquitin-specific protease-14 (USP14) by Akt activates USP14 DUB activity. (**A**) Akt activates USP14 DUB activity in vitro. USP14 protein (1 μg) was incubated with or without active Akt (1 μg) in kinase assay buffer in a total volume of 50 μL for 1 hr at 30$^\circ$C, then the reaction mixtures were subjected to Ub-AMC assay. RFU, relative fluorescence units. (**B, C**) Akt activates USP14 in cells. USP14 was immunoprecipitated from HEK293T cells coexpressed with activated Akt (**B**) or treated with 10 μM MK2206 for 4 hr (**C**) and then eluted with HA-peptide following Ub-AMC hydrolysis assay. (**D**) Activation of USP14 by stimulating cells with IGF-1. HEK293T cells were serum-starved and pretreated with or without Akt inhibitor MK2206 (1 μM) for 30 min before stimulation with IGF-1 (100 ng/mL) for 30 min. USP14 was then immunoprecipitated and eluted with HA-peptide. The activity of USP14 was determined using Ub-AMC hydrolysis assay. (**E**) USP14 activation by Akt is blocked by S432A mutation. Ub-AMC hydrolysis assay of wildde type USP14 or S432A mutant in the presence or absence of active Akt. (**F**) Ub-AMC hydrolysis assay of bacterially expressed and purified wild type USP14 or S432E mutant. (**G**) Lineweaver–Burk analysis of USP14 S432E, obtained by measuring the initial rates at varying Ub-AMC concentrations (see *Figure 3—figure supplement 2E* for reference). (**H**) The activity of phospho-mimetic USP14 mutant can be further stimulated by the presence of proteasome. Ub-AMC hydrolysis assay of wild type USP14 or S432E mutant in the presence or absence of Ub-VS-treated human proteasome (VS-proteasome (see *Lee et al., 2010*); 1 nM). Ptsm, 26S proteasome.

The following figure supplements are available for figure 3:

**Figure supplement 1.** Purification of ubiquitin-specific protease-14 (USP14) recombinant protein.

*Figure 3 continued*

**Figure supplement 2.** Phosphorylation of USP14 by Akt activates USP14 DUB activity.

(*Figure 4—figure supplement 1B,C*). In contrast, S432A mutant immunoprecipitated from cells showed lower activity towards both Lys48 and Lys63 diubiquitin than that of WT (*Figure 4C*).

## Regulation of UPS by Akt depends on phosphorylation of USP14

Since USP14 is a negative regulator of the UPS (*Koulich et al., 2008*; *Lee et al., 2010*, *2011*) and we found USP14 can be phosphorylated and activated by Akt, we reasoned that Akt-mediated activation of USP14 might lead to inhibition of the UPS and generally enhance the stability of many proteins. To this end, we generated a stable cell line expressing GFP-CL1 (also known as GFPu), an engineered ubiquitin-dependent proteasome substrate widely used as a reporter for UPS activity (*Bence et al., 2001*; *Kelly et al., 2007*; *Li et al., 2013*; *Liu et al., 2014*) (*Figure 5—figure supplement 1A–C*). Treatment of cells with Akt inhibitors or serum deprivation or PI3K inhibitor, all of which can block Akt activity (*Zhang et al., 2015*), led to reduced level of GFP-CL1 as detected by both western blotting and fluorescence microscopy (*Figure 5A–C* and *Figure 5—figure supplement 1D*). Conversely, the expression of activated Akt (Myr-Akt) led to increased levels of GFP-CL1 protein. Treatment of WT H4 cells with IGF-1 or EGF also led to increased levels of GFP-CL1 protein (*Figure 5D–G* and *Figure 5—figure supplement 1E*). In contrast, in USP14 knockout H4 cells (generated using CRISPR/Cas9 technology, *Figure 5—figure supplement 2A–D*), the expression of Myr-Akt did not affect the levels of GFP-CL1 (*Figure 5H*). From these results, we conclude that Akt negatively regulates the UPS in an USP14-dependent manner.

We next tested the importance of USP14 phosphorylation for Akt to regulate UPS. We found that in contrast to USP14 WT reconstituted H4 cells, USP14 AA mutant reconstituted H4 cells showed no increase in the accumulation of GFP-CL1 in response to the expression of activated Akt (*Figure 5—figure supplement 2E*and *Figure 5I*). As a control, we found that the expression of Akt had no effect on a ubiquitin-independent substrate of the proteasome, C-terminal ornithine decarboxylase-GFP (GFP-cODC) (*Hoyt et al., 2005*; *Kelly et al., 2007*; *Lee et al., 2010*) (*Figure 5—figure supplement 2F,G*), suggesting that Akt does not inhibit the UPS through a general inhibition of the proteasome itself. Taken together, these data show that phosphorylation of USP14 by Akt is important for this kinase to negatively regulate the UPS in a ubiquitin-dependent manner.

## Phosphorylation of USP14 regulates global protein degradation

To further understand the physiological roles of Akt-mediated USP14 phosphorylation and subsequently activation, we sought to study the impact of USP14 phosphorylation on global protein degradation. Since the loss of USP14 accelerates cellular proteolysis (*Koulich et al., 2008*; *Lee et al., 2010*), we performed a quantitative proteomic analysis to determine the levels of proteins in WT H4 cells, H4 USP14-KO cells, and H4 USP14-KO cells complemented with WT USP14, S143A/S432A (AA), or S143D/S432D (DD) mutants. Using an isobaric tandem mass tag (TMT) labeling approach, our mass spectrometry analysis identified 18,400 peptides with high confidence (q<0.01), corresponding to 3,648 proteins with a minimum of two peptides from each protein. A total of 2,763 proteins, which were quantified in at least two replicates, were subjected to further analysis. We found the global protein patterns of H4 USP14-KO cells were similar to those of H4 USP14 KO-AA cells, but distinct from those of WT H4 cells. We identified a common set of 87 proteins that were reduced in H4 KO cells as compared to H4 WT cells or to H4 KO cells complemented with WT USP14 (KO-WT) (*Figure 6*, Lane1-2). The levels of these proteins were also significantly reduced in H4 KO-AA cells (*Figure 6*, Lane 3). Importantly, the levels of this set of 87 proteins in H4 KO-DD cells were significantly higher than that of H4 KO-AA cells (*Figure 6*, Lane 4).

To verify that the identified changes in protein abundance were due to proteasomal degradation, we treated H4 KO-AA cells with proteasome inhibitor MG132 and analyzed the protein level change of these 87 proteins. We found that the levels of these proteins increased significantly in MG132-

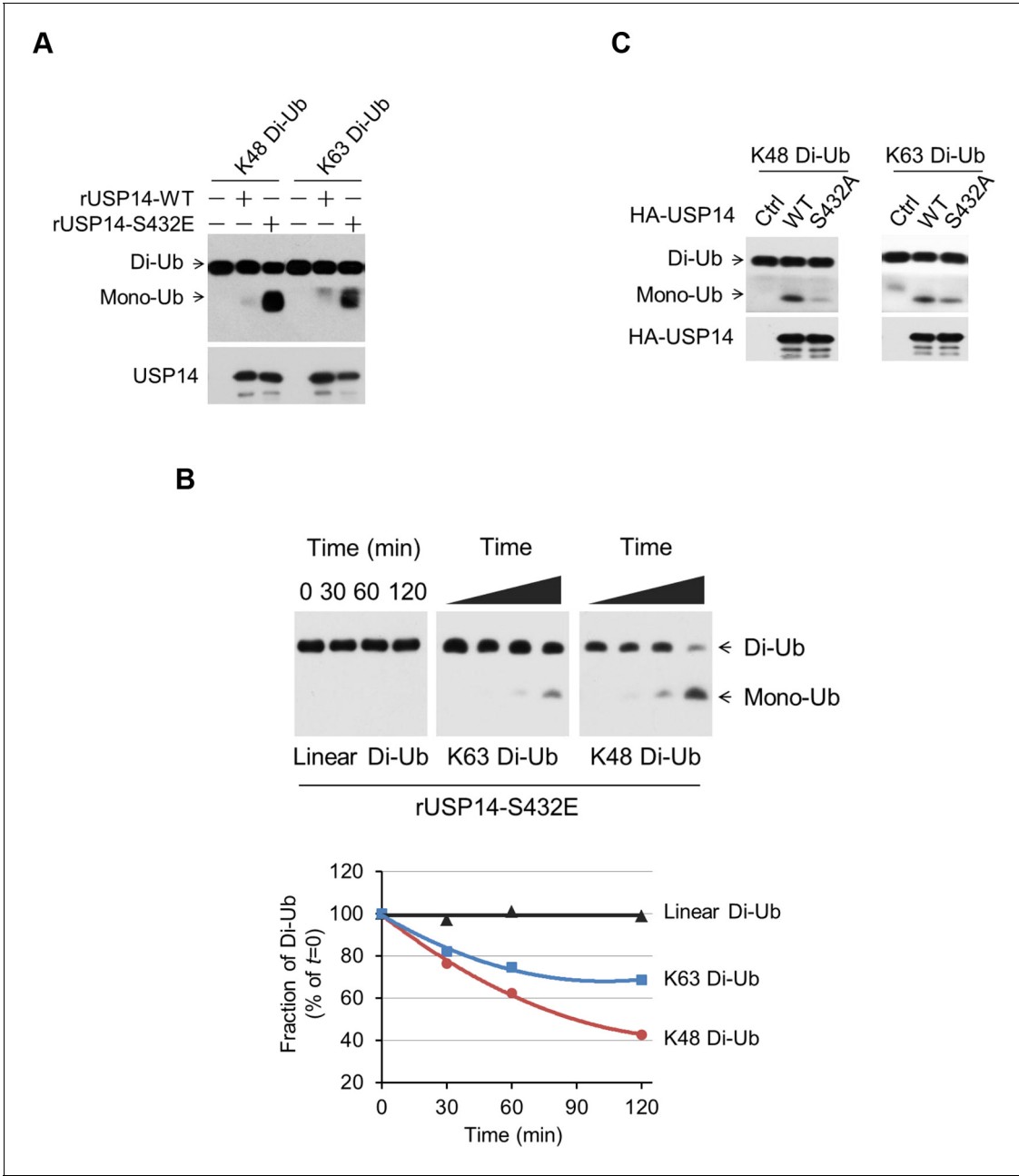

**Figure 4.** Phosphorylation stimulates ubiquitin-specific protease-14 (USP14) activity towards both K48 and K63 ubiquitination. (**A**) Dimeric Ub cleavage assay. USP14 S432E cleavage of Lys 48 and Lys 63 Ub chain linkages was analyzed using sodium dodecyl sulfate polyacrylamide gel electrophoresis (SDS-PAGE). (**B**) Cleavage of Lys 48, Lys 63 and linear dimeric Ub chain types in the presence of USP14 S432E was measured over time and analyzed using SDS-PAGE. Quantification of the amount of dimer remaining from the data was shown below. (**C**) Dimeric Ub cleavage analysis of immunoprecipitates derived from HEK293T cells transfected with wild type HA-USP14 or S432A mutant plasmids.

The following figure supplement is available for figure 4:

**Figure supplement 1.** Phosphorylation of ubiquitin-specific protease-14 (USP14) promotes both K48 and K63 deubiquitination activity.

treated KO-AA cells compared to that of control KO-AA cells (*Figure 6*, Lane 5), suggesting that these proteins were indeed subject to an increased rate of proteasome degradation with expression of non-phosphorylatable USP14. Interestingly, the top hit on this list of 87 proteins that were differentially regulated upon the loss of USP14 is mTOR, a central established regulator of cellular

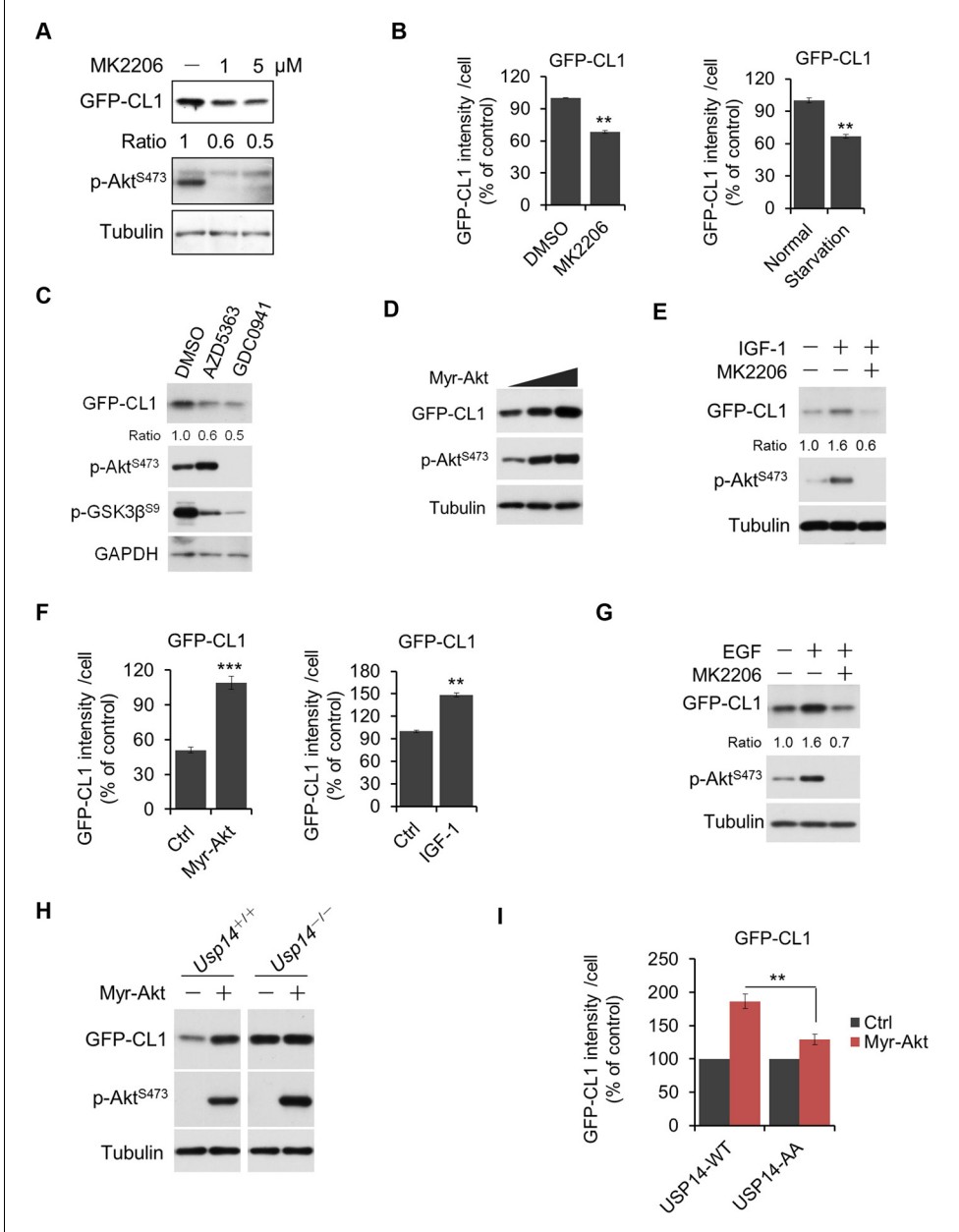

**Figure 5.** Akt regulates ubiquitin–proteasome system (UPS) function through phosphorylation of ubiquitin-specific protease-14 (USP14). (**A**) Inhibition of Akt promotes UPS function. H4-GFP-CL1 cells were treated with different concentrations of MK2206 as indicated for 4 hr. The cells were then harvested and subjected to western blotting analysis using indicated antibodies. (**B**) H4-GFP-CL1 cells were treated as in (**A**) and *Figure 5—figure supplement 1D*. Images of the cells were collected using an ArrayScan HCS 4.0 Reader. The average GFP intensity in 2,000 cells from each indicated sample was determined. Data are displayed as mean ± SD of the GFP intensity per cell. **$p<0.01$, ***$p<0.001$ (**C**) Inhibition of Akt or phosphoinositide 3-kinase (PI3K) promotes UPS function. H4-GFP-CL1 cells were treated with Akt inhibitor AZD5363 (1 µM) or PI3K inhibitor GDC0941 (1 µM) as indicated for 4 hr. The cells were then harvested and subjected to western blotting analysis using indicated antibodies. p-GSK3β(S9) was blotted to indicate the inhibition of Akt. (**D**) Activation of Akt inhibits UPS. H4-GFP-CL1 cells were transfected with Myr-Akt for 24 hr. The cells were then harvested and subjected to western blotting analysis using indicated antibodies. (**E**) IGF-1 stimulation inhibits UPS function. H4-GFP-CL1 cells were serum-starved and pretreated with Akt inhibitor MK2206 (1 µM) for 30 min before stimulating with IGF-1 (100 ng/mL) for 30 min. The cells were then imaged and quantified as in (**B**). (**F**) H4-GFP-CL1 cells were treated as in (**D**) and *Figure 5—figure supplement 1E*. Then cells were imaged and quantified as in (**B**). (**G**) EGF stimulation inhibits UPS function. H4-GFP-CL1 cells were serum-starved and pretreated with Akt inhibitor MK2206 (1 µM) for 30 min before stimulation with EGF (100 ng/mL) for 1 h. The cells were then harvested and subjected to western blotting analysis using indicated antibodies. (**H**) Akt regulates UPS function through USP14. Myr-Akt was transfected into either wild type or *Usp14⁻/⁻* H4 cells stably expressing GFP-CL1 for 24 hr. The cells were then harvested and subjected to western blotting analysis using indicated antibodies. (**I**) Akt regulates UPS function through phosphorylation of USP14. Myr-Akt was

*Figure 5 continued on next page*

*Figure 5 continued*

transfected into either wild type USP14 or USP14 AA reconstitution cell lines stably expressing GFP-CL1 for 24 hr. The cells were then imaged and quantified as in (B).

The following figure supplements are available for figure 5:

**Figure supplement 1.** Regulation of UPS by Akt.

**Figure supplement 2.** Regulation of UPS by Akt depends on phosphorylation of USP14.

metabolism and tumorigenesis. We confirmed the role of USP14 on the levels of mTOR by western blotting. We found that the levels of mTOR were reduced in H4 KO and H4 KO cells complemented with USP14 AA mutant, but restored upon the expression of USP14 DD mutant (*Figure 6—figure supplement 1*). Taken together, our results suggest that phosphorylation of USP14 may provide a mechanism for Akt to regulate global protein degradation through the proteasome, which in turn may control key cellular pathways involved in regulating metabolism and tumorigenesis.

## Discussion

We report here a new mode of USP14 regulation involving its phosphorylation by the protein kinase Akt. The activity of USP14 is induced 800-fold by association with the proteasome (*Lee et al., 2010*). Remarkably, Akt-dependent phosphorylation of USP14 elevates the catalytic activity of proteasome-associated USP14 beyond this level. Thus, Akt not only activates USP14 by a different mechanism than the proteasome, but it can cooperate with the proteasome to achieve more aggressive removal of ubiquitin from proteasome-docked substrates. Akt activation of USP14 has marked effects on protein turnover in cells, highlighting the physiological significance of the activity state of USP14. USP14 and its ortholog from yeast (Ubp6) can suppress degradation through both deubiquitination and a noncatalytic activity (*Bashore et al., 2015*; *Hanna et al., 2007*; *Lee et al., 2010*). The ability to rapidly adjust the rate of proteasomal degradation by responding to external signals, including growth factors, nutritional demands, and stress, is critical for maintaining cellular survival and preventing the toxicity associated with protein misfolding and accumulation of toxic aggregates. By regulating the rate of proteasomal degradation through phosphorylating USP14, Akt may exert controls over the entire proteome of short-lived proteins as well as cellular protein quality.

Since Akt is dramatically activated in PTEN-deficient cancer cells, the control of USP14 phosphorylation by Akt may provide a mechanism for cancer cells with PTEN loss, one of the most common cancer mutations, to control global intracellular proteostasis by regulating protein degradation through proteasomes. Furthermore, since Akt can be activated by a wide range of growth factors, such as insulin, EGF, IGF, and fibroblast growth factor (FGF), through their respective receptors, regulation of USP14 by Akt-mediated phosphorylation may provide a general mechanism for growth factors to control global proteostasis during cell growth. Our results do not exclude the possibility that other kinases that are activatable by growth factors and activated in PTEN-deficient conditions can also mediate the phosphorylation of USP14.

The precise control of the UPS allows timely and selective degradation of surplus and/or aberrant proteins which is essential for normal cellular physiology. Dys-regulation of UPS may disrupt cellular proteostasis and lead to the inappropriate accumulation of target proteins to compromise cellular and tissue homeostasis. Since UPS is critically involved in the degradation of cellular short-lived proteins which frequently serve in mediating intracellular signaling process, the regulation of UPS by Akt-mediated phosphorylation of USP14 may provide a mechanism to control multiple signaling process, raising the possibility that control of short-lived proteins might serve as an active process that has impact on cellular signaling, rather than as a degradative process per se. In this regard, regulation of UPS by Akt has been proposed to contribute to type II diabetes mediated by inflammation. Stimulation of adipocytes by TNFα has been shown to lead to reduction of Akt (*Medina et al., 2005*). Thus, a reduction of Akt might promote UPS to lead to global proteomic changes to contribute to the pathological consequence in diabetes.

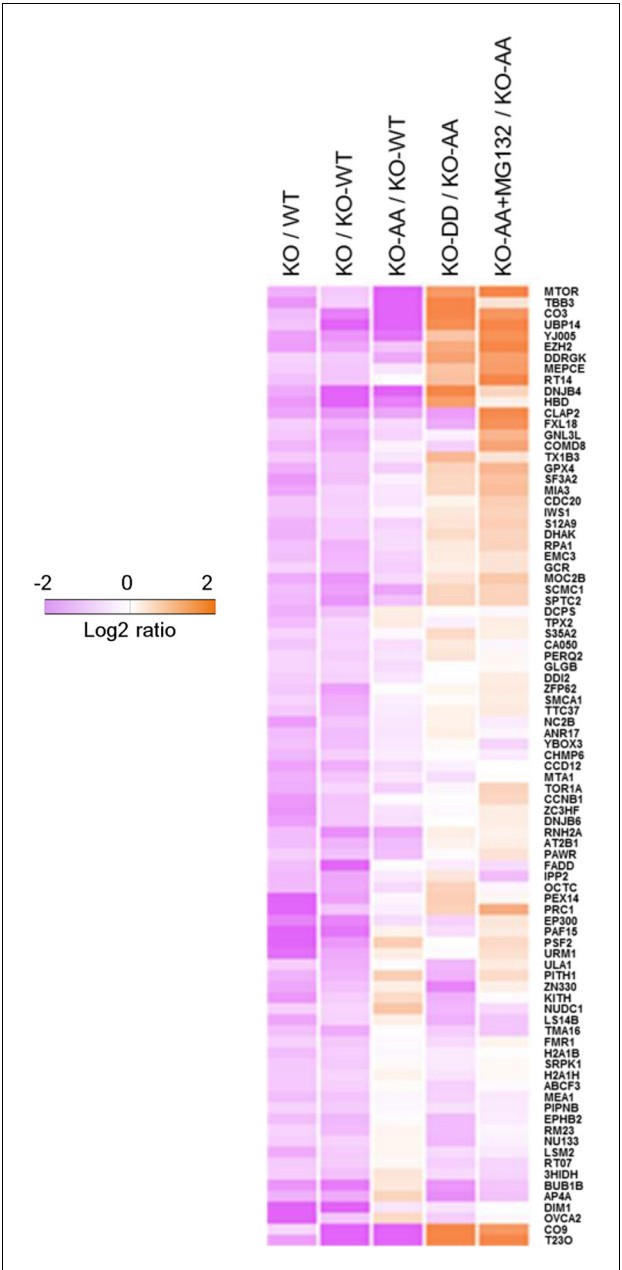

**Figure 6.** Phosphorylation of ubiquitin-specific protease-14 (USP14) regulates global protein degradation. The quantitative analysis of proteome change in USP14 knockout or USP14 mutant cells were performed by tandem mass tag (TMT)-isobaric labeling followed by shotgun analysis. The heat map was plotted based on the set of 87 proteins that are down-regulated greater than or equal to 1.2-fold in H4 KO cells compared to H4 WT cells or to H4 KO cells complemented with WT USP14 (KO-WT). The log base 2 of average ratios was plotted as indicated.

The following figure supplement is available for figure 6:

**Figure supplement 1.** Western blotting analysis of mTOR expression in WT cells and USP14 mutant-reconstituted cells.

# Materials and methods

## Cell culture

All cell lines were maintained at 37°C with 5% $CO_2$. HEK293T cells were cultured in DMEM (Gibco) with 10% (vol/vol) FBS (Gibco) and 1% penicillin/streptomycin. H4 and H4-$Usp14^{-/-}$ cells were maintained in DMEM supplemented with 10% (vol/vol) FBS, 1% penicillin/streptomycin, and 1×sodium pyruvate (Invitrogen).

## Antibodies and reagents

Commercial antibodies used for western blotting analysis include: anti-Phospho-Akt (Ser473) (#3787, 1:1000 dilution), anti-Phospho-Akt Substrate (R××pS/pT) (#9614, 1:1000 dilution for WB and 1:250 dilution for IP), anti-USP14 (rabbit, #11931, 1:1000 dilution), anti-FLAG (#2368, 1:1000 dilution) were from Cell Signaling Technology. Anti-phosphoserine (#61-8100, 1:1000 dilution for WB and 1:250 dilution for IP) was from Invitrogen. Anti-Akt (sc-8312, 1:1000 dilution) and anti-USP14 (mouse, sc-393872, 1:1000 dilution) were from Santa Cruz Biotechnology. Anti-Myc (16286-1-AP, 1:1000 dilution), anti-HA (51064-2-AP, 1:1000 dilution), and anti-GAPDH (60004-1-Ig, 1:10000 dilution) were from Proteintech. Anti-Tubulin (PM054, 1: 10000 dilution) was from MBL. Recombinant active Akt protein (#14-276) was from Millipore. Mouse monoclonal Anti-FLAG (F1804, 1:500 for IP), anti-Myc (A7470), and anti-HA Affinity Gel (E6779) were from Sigma-Aldrich. IU1 (S7134), MK2206 (S1078), AZD5363 (S8019), GDC0941(S1065), Wortmannin (S2758), U0126 (S1102), and MG132 (S2619) were from Selleckchem. Phospho-USP14 S432 antibodies were generated by Proteintech. Briefly, synthetic peptides corresponding to phosphorylated S432 epitope (N'-Cys-THQGRSSs(phospho)SGHYVSW) of USP14 were used to immunize rabbits and the antisera were affinity purified.

## Plasmid transfection

The cDNAs for USP14 and a constitutively active form of Akt, N-terminally myristoylation signal (MG-SSKSKPK)-attached Akt (Myr-Akt), were cloned into pcDNA3.1 using Phanta Max Super-Fidelity DNA Polymerase (Vazyme Biotech Co., Ltd) and ClonExpress$^{TM}$ II cloning kit (Vazyme Biotech Co., Ltd). GFP-CL1 was derived by adding the CL1 degron (ACKNWFSSLSHFVIHL) to the C-terminal of GFP and cloned into pMSCV to generate retroviral transfer vector. Mutagenesis was performed using MutExpress$^{TM}$ II mutagenesis kit (Vazyme Biotech Co., Ltd). The cells were transfected with plasmid DNA using PolyJet DNA In Vitro Transfection Reagent (Signagen Laboratories) according to the manufacturer's instructions.

## Protein expression and purification

The USP14 wild type or mutant DNA was cloned into pET-32M vector (an in-house modified version of pET32a vector containing a N-terminal Trx-tag and His$_6$-tag). Recombinant proteins were expressed in BL21 (DE3) *E. coli* cells. The bacterial cultures were grown at 37°C until $OD_{600\ nm}$ reached 0.6–0.8, and USP14 expression was then induced overnight with 0.2 mM IPTG at 16°C. The cells were harvested in binding buffer (50 mM Tris-HCl (pH 7.5), 500 mM NaCl, 5 mM imidazole) containing protease inhibitors and lysed by the NANO homogenizer machine (FBE, Shanghai). The lysate was then clarified by centrifugation at 18,000 × *g* for 30 min. His$_6$-tagged proteins were purified by Ni$^{2+}$-NTA agarose (Qiagen) affinity chromatography. Each recombinant protein was further purified by size-exclusion chromatography. The terminal tag of each recombinant protein was cleaved by 3C protease overnight at 4°C and further removed by size-exclusion chromatography.

## In vitro kinase assay

Recombinant USP14 or USP14 mutant protein (1 µg) was incubated with 1 µg active Akt, 0.2 mM ATP, and kinase assay buffer (Cell Signaling) in a total volume of 50 µl for 1 hr at 30°C. The reaction mixtures were subjected to Ub-AMC assay by the addition of 50 µl 2×Ub-AMC buffer. Alternatively, the kinase reaction was stopped by the addition of 50 µl 2×sample buffer, and resolved by SDS-PAGE, followed by blotting with phospho-specific antibodies.

## Glycerol density gradient centrifugation

*Pten*$^{-/-}$ MEFs cells were lysed in buffer A (20 mM Tris-HCl (pH 7.6), 20 mM NaCl, 1 mM β-mercaptoethanol, 1 mM ATP, and 5 mM MgCl$_2$). After 10 min at 4°C, cells were disrupted with 50 passages through a 27-gauge needle. Lysates were centrifuged at 16,000 × *g* for 10 min, supernatants were supplemented with 10% glycerol. Density gradient centrifugation was conducted in 10–40% linear glycerol gradients. Gradients contained 50 mM Tris-HCl (pH 7.6), 20 mM NaCl, 1 mM dithiothreitol, 1 mM ATP, and 5 mM MgCl$_2$. Samples were centrifuged at 55,000 × *g* for 3 hr. Fractions were collected for further analysis.

## Generation of UPS reporter lines, reconstitution lines, and USP14 knockout cells

For UPS reporter cell line, H4 cells and *Usp14*$^{-/-}$ H4 cells were infected with retroviral particles expressing GFP-CL1 or GFP-cODC, and then selected with 1 µg/ml puromycin to generate stable cell lines. For reconstitution lines, *Usp14*$^{-/-}$ H4 cells were infected with lentiviral particles expressing HA-USP14 (WT or mutant). *Usp14* was knocked out from H4 cells using the CRISPR/Cas9 system (*Jinek et al., 2013*), with a guide RNA spanning exon 2. The guide RNA was individually cloned into the pX330 vector and transfected into H4 cells. Transfected cells were sorted by fluorescence-activated cell sorting using green fluorescent protein. Single colonies were screened using PCR to confirm the expected genomic deletion and western blot to confirm the loss of USP14 protein expression.

## Ub-AMC and ubiquitin cleavage assay

Ub AMC-conjugated proteins were purchased from Boston Biochem. Assays were carried out in a flat-bottom, low-flange 384-well plate in a 40 µl reaction. Enzymes and substrates were prepared in Ub-AMC assay buffer (50 mM Tris-HCl (pH 7.5), 1 mM EDTA, 1 mM ATP, 5 mM MgCl$_2$, 1 mM DTT, and 1 mg/ml ovalbumin). The reaction was initiated by adding of Ub-AMC and measured at Ex345/Em445 using an Envision plate reader (PerkinElmer). For determination of $K_M$, USP14-S432E was incubated with the indicated concentrations of Ub-AMC. Lineweaver–Burk analysis was carried out using a linear regression fit of the data with the equation

$$\left(\frac{1}{\nu}\right) = \frac{K_M}{V_{\max}}\left(\frac{1}{[S]}\right) + \frac{1}{V_{\max}}$$

to calculate $K_M$.

For ubiquitin cleavage assays, hydrolysis reactions were carried out at 30°C in reaction buffer, with a constant enzyme concentration of 400 nM USP14 and USP14 S432E in a 50 µl reaction volume. Aliquots of 10 µl were taken at the indicated times and added to 10 µl SDS loading buffer to stop the reaction.

## Mass spectrometry and data analysis

The USP14 protein was subject to trypsin digestion on beads after immunoprecipitation experiment. The enrichment of phosphorylated peptides by TiO$_2$ was performed with tryptic USP14 peptides. The enriched phosphorylated peptides were analyzed on Orbitrap Fusion mass spectrometer (Thermo Scientific). The activation type of HCD was performed for MS2. Protein identification was performed by Thermo Proteome discoverer (v1.4) with Sequest HT. The precursor mass tolerance was set at 10 ppm, and the fragment mass tolerance was set at 0.1 Da. The cysteine carboxyamido methylation was set as a static modification, and phosphorylated serine, threonine, and tyrosine residues were set as variable modifications. The peptide false-positive rate was controlled to be less than 1%. PhosphoRS site probability analysis was performed.

Quantitative analysis of proteome changes in USP14 knockout or USP14 mutant cells was performed by TMT-isobaric labeling followed by shotgun analysis. The cell lysate of WT, *Usp14*$^{-/-}$, *Usp14*$^{-/-}$ reconstituted with WT USP14 (KO-WT), *Usp14*$^{-/-}$ reconstituted with AA mutant USP14 (KO-AA), *Usp14*$^{-/-}$ reconstituted with DD mutant USP14 (KO-DD), and KO-AA cells treated with 10 µM MG132 for 4 hr were digested with trypsin and labeled with 126, 127, 128, 129, 130, 131-TMT labeling reagent (Thermo Scientific), respectively, according to the manufacturer's instruction. Equal amount of peptides with each TMT tag were mixed, and the resulting mixture of peptides was subjected to fractionation using off-line high pH reverse phase chromatography. Six fractions were

collected and subsequently analyzed on Orbitrap Fusion mass spectrometer. Three replicates were performed. Protein identification and quantification was done by Thermo Proteome discoverer (v1.4). The peptide false-positive rate was controlled to be less than 1%. The peak integration tolerance was set at 10 ppm. Only unique peptides were used for protein quantitation. An isobaric tag purity correction was performed. The labeling efficiency was measured and was greater than 99%. The quantitation normalization based on protein median was performed. The average ratios of each protein from three replicates were used for analysis.

### Cell imaging and statistical analysis

Cells were fixed with 4% paraformaldehyde and stained with 3 µg/ml DAPI (Sigma). Images data were collected with an ArrayScan HCS 4.0 Reader with a 203 objective (Cellomics ArrayScan VTI) for DAPI-labeled nuclei and GFP-tagged intracellular proteins. Error bars for microscopy were presented as the standard deviation of triplicate samples. Error bars for Western blotting analysis represent the standard deviation between densitometry data from three biological replicates. Student's t-test was used as statistical analysis by using GraphPad Prism.

## Acknowledgements

PTEN-WT and knockout MEFs cells were kindly provided by Dr. Hong Wu of Peking University. This work was supported in part by grants (to JY) from the Chinese Academy of Sciences, from the National Institute on Aging (US) (1R01AG047231), the NINDS (US) (1R01NS082257) (to JY), and from the Global Research Laboratory Program (GRL, NRF-2010-00341) by the Ministry of Science and Technology (MEST) in Korea, and by NIH grant 5R01GM095526 (to DF).

## Additional information

### Funding

| Funder | Grant reference number | Author |
| --- | --- | --- |
| National Institute on Aging | 1R01AG047231 | Junying Yuan |
| National Institute of Neurological Disorders and Stroke | 1R01NS082257 | Junying Yuan |
| Chinese Academy of Sciences | | Junying Yuan |
| National Institutes of Health | 5R01GM095526 | Daniel Finley |
| Ministry of Science, ICT and Future Planning | Global Research Laboratory Program, GRL, NRF-2010-00341 | Junying Yuan |

The funders had no role in study design, data collection and interpretation, or the decision to submit the work for publication.

### Author contributions

DX, Conception and design, Acquisition of data, Analysis and interpretation of data, Drafting or revising the article, Contributed unpublished essential data or reagents; BS, LP, Acquisition of data, Analysis and interpretation of data, Drafting or revising the article, Contributed unpublished essential data or reagents; BHL, KZ, TZ, HS, ML, LS, WL, LQ, JX, LW, Acquisition of data, Analysis and interpretation of data, Contributed unpublished essential data or reagents; DF, Conception and design, Drafting or revising the article, Contributed unpublished essential data or reagents; JY, Conception and design, Analysis and interpretation of data, Drafting or revising the article, Contributed unpublished essential data or reagents

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
