## [Decision Letter]

Thank you for submitting your work entitled "Phosphorylation and activation of ubiquitin-specific protease-14 by Akt regulates UPS" for peer review at *eLife*. Your submission has been favorably evaluated by Randy Schekman (Senior editor) and two reviewers, one of whom is a member of our Board of Reviewing Editors.

The reviewers have discussed the reviews with one another and the Reviewing editor has drafted this decision to help you prepare a revised submission.

Xu et al. report that the protein kinase AKT phosphorylates the deubiquitinase USP14 at Ser432 and Ser143 in vitro and in overexpression experiments, and that the phosphorylation of Ser432 is accompanied by a large increase in USP14 catalytic activity. They also report that USP14 undergoes phosphorylation at Ser432 in cells, that this is decreased by serum deprivation, increased in response to stimulation with IGF1 and suppressed by MK2206, which prevents the phosphorylation and hence the activation of AKT by the protein kinases PDK1 and TORC2. They further show that USP14 is phosphorylated constitutively at Ser432 in PTEN null cells. Finally they report that the phosphorylation of USP14 suppresses the activity of the proteasome in cells.

1) The data presented in the paper provide convincing evidence that the phosphorylation of USP14 at Ser432, and perhaps also Ser143, activates USP14 activity in vitro and in overexpression experiments in cells. However, data presented on endogenous USP14 and the endogenous kinase is limited (Figure 2). These data need to be expanded and the quality of the data needs to be improved. First, an IP-western should also be performed to show that the band detected is USP14 and not a different protein of similar molecular weight. Second, the experiments should be performed using cells treated with MK2206 without and with IGF1-treatment and also with a different activator of AKT (e.g. EGF). Third, the concentration of MK2206 should be stated in the legend – in other Figures the range is 1-10µM; here it is important that a low dose (e.g. 1 µM) is used to avoid non-specific artefacts. Fourth, a second AKT inhibitor structurally unrelated to MK2206 (e.g. AZD5363) should be used. Fifth, suppression of phosphorylation by PI 3-kinase inhibitors (Wortmannin, GDC0941) should be presented. Sixth, the effect of a MEK inhibitor (to suppress activation of RSKs) should be presented.

2) Figure 2 reports that the endogenous USP14 is phosphorylated in PTEN KO cells in the absence of IGF1 stimulation. In addition to strengthening these data in the same way suggested for the result in Figure 2, it is critical to show that phosphorylation is suppressed by MK2206 to be able to conclude that phosphorylation is catalyzed by AKT. Moreover, AKT is not the only protein kinase that is activated downstream of PI 3-kinase. The protein kinases SGK1, SGK2 and SGK3, which are the most similar protein kinases in the human genome to AKT1, 2 and 3, have a similar substrate specificity to AKT and, like AKT, are activated by PDK1 and TORC2. They are therefore candidates to phosphorylate USP14, but their activation is not blocked by MK2206. Protein kinases of the RSK family are also activated by stimulation with growth factors and deactivated by serum starvation and they also have similar substrate specificities to AKT. The consensus sequence for phosphorylation by AKT, as well as SGKs and RSKs is not RxxS as stated in the paper, but RxRxxS and most of the known physiological substrates of these kinases are phosphorylated at this motif. All protein kinases of the AGC and CAMK subfamilies as well as other kinases phosphorylate RxxS sequences. So about 20% of all protein kinases phosphorylate this motif. Ser432 and Ser143 do not lie in RxRxxS sequences, which is a further reason to be cautious.

3) The authors point out in the Introduction that USP14 can be activated in vitro by interaction with the proteasome. How does this relate to the observation that USP14 is activated by AKT? The phos-tag gels suggest that only a minor fraction of USP14 is phosphorylated. Is it only the proteasome bound fraction of USP14 that is phosphorylated or is it the unbound USP14 that is phosphorylated. Can USP14 activated by the proteasome be further activated by phosphorylation?

4) Figure 3: To be sure that it is HA-USP14 and not another DUB contaminating the IPs that is being measured, it is essential to perform a control experiment in which a catalytically inactive USP14 mutant is used instead of the wild type protein.

5) In Figure 3, the effects of MK2206 on USP14 activity seem modest (compared with the effects on USP14 phosphorylation presented elsewhere).

6) In Figure 3, the S432A mutant is still activated by AKT, suggesting that Ser143 phosphorylation may cause partial activation. An experiment with Ser143Ala and Ser143Glu mutants would be informative.

7) Can USP14 activation be observed by stimulating with IGF1 in the absence of any overexpression of AKT, and if so, is it suppressed by MK2206 – this is a more critical experiment.

8) Figure 5: Does IGF1 stimulation without AKT overexpression activate the UPS and, if so, is this blocked by MK2206?

---

## [Author Response]

*1) The data presented in the paper provide convincing evidence that the phosphorylation of USP14 at Ser432, and perhaps also Ser143, activates USP14 activity in vitro and in overexpression experiments in cells. However, data presented on endogenous USP14 and the endogenous kinase is limited (Figure 2). These data need to be expanded and the quality of the data needs to be improved. First, an IP-western should also be performed to show that the band detected is USP14 and not a different protein of similar molecular weight.*

We thank the reviewers for this suggestion, which helped to improve our manuscript. Indeed, we have immunoprecipitated USP14 and then western blotting for p-S432 ab to confirm that it is USP14 that is detected by this p-S432 ab (Figure 2, Figure 2—figure supplement 1).

*Second, the experiments should be performed using cells treated with MK2206 without and with IGF1-treatment and also with a different activator of AKT (e.g. EGF).*

As requested, we performed the experiments using cells treated with or without MK2206 (Figure 2—figure supplement 1) and with IGF1-treatment (Figure 2, Figure 5) and also with a different activator of AKT, EGF (Figure 2, Figure 5).

*Third, the concentration of MK2206 should be stated in the legend – in other Figures the range is 1-10µM; here it is important that a low dose (e.g. 1 µM) is used to avoid non-specific artefacts.*

Now the concentrations of MK2206 used are stated in the legends.

*Fourth, a second AKT inhibitor structurally unrelated to MK2206 (e.g. AZD5363) should be used.*

As requested, we tested AZD5363 (Figure 2—figure supplement 1, Figure 5). p-GSK3β(S9), a known Akt substrate (Zhang et al., 2015), was detected by WB to indicate the inhibition of Akt.

*Fifth, suppression of phosphorylation by PI 3-kinase inhibitors (Wortmannin, GDC0941) should be presented.*

As requested, PI3K inhibitors GDC0941 and Wortmannin were tested (Figure 2—figure supplement 1, Figure 5).

*Sixth, the effect of a MEK inhibitor (to suppress activation of RSKs) should be presented.*

As requested, inhibition of ERK1/2 by U0126 was tested (Figure 2—figure supplement 1).

*2) Figure 2 reports that the endogenous USP14 is phosphorylated in PTEN KO cells in the absence of IGF1 stimulation. In addition to strengthening these data in the same way suggested for the result in Figure 2, it is critical to show that phosphorylation is suppressed by MK2206 to be able to conclude that phosphorylation is catalyzed by AKT.*

As requested, this experiment was performed to show that phosphorylation of USP14 in PTEN KO cells was indeed suppressed by MK2206 and AZD5363 (Figure 2—figure supplement 1).

*Moreover, AKT is not the only protein kinase that is activated downstream of PI 3-kinase. The protein kinases SGK1, SGK2 and SGK3, which are the most similar protein kinases in the human genome to AKT1, 2 and 3, have a similar substrate specificity to AKT and, like AKT, are activated by PDK1 and TORC2. They are therefore candidates to phosphorylate USP14, but their activation is not blocked by MK2206. Protein kinases of the RSK family are also activated by stimulation with growth factors and deactivated by serum starvation and they also have similar substrate specificities to AKT. The consensus sequence for phosphorylation by AKT, as well as SGKs and RSKs is not RxxS as stated in the paper, but RxRxxS and most of the known physiological substrates of these kinases are phosphorylated at this motif. All protein kinases of the AGC and CAMK subfamilies as well as other kinases phosphorylate RxxS sequences. So about 20% of all protein kinases phosphorylate this motif. Ser432 and Ser143 do not lie in RxRxxS sequences, which is a further reason to be cautious.*

We thank the reviewers for this comment. We have added a sentence to the end of second paragraph in Discussion: “Our results do not exclude the possibility that other kinases that are activatable by growth factors and activated in PTEN deficient conditions can also mediate the phosphorylation of USP14”.

*3) The authors point out in the Introduction that USP14 can be activated in vitro by interaction with the proteasome. How does this relate to the observation that USP14 is activated by AKT?*

To address this question, we have conducted two experiments. 1) We analyzed the distribution of phosphorylated and total USP14 by glycerol gradient centrifugation (Koulich et al., 2008) and found that phosphorylated USP14 was largely present in fractions without proteasome while unphosphorylated USP14 was found in the same fractions with proteasome (Figure 3—figure supplement 2). 2) We determined the Ub-AMC hydrolysis activity of WT and S432E mutant USP14 protein in the presence of proteasome in vitro and found that the activity of S432E can be further stimulated by proteasome (Figure 3).

*The phos-tag gels suggest that only a minor fraction of USP14 is phosphorylated. Is it only the proteasome bound fraction of USP14 that is phosphorylated or is it the unbound USP14 that is phosphorylated.*

To address this question from the reviewers, we performed glycerol density gradient centrifugation (Koulich et al., 2008) to determine the distribution of free and proteasome-associated phosphorylated USP14 in *Pten^-/-^*MEFs (with high USP14 S432 phosphorylation) cells lysates (Figure 3—figure supplement 2). The phos-USP14 is preferentially distributed in fractions without proteasome.

*Can USP14 activated by the proteasome be further activated by phosphorylation?*

To address this question, we have analyzed the activity of S432E mutant and WT in the presence of proteasome and found that the activity of phospho-mimetic USP14 mutant can be further activated by the proteasome (Figure 3).

*4) Figure 3: To be sure that it is HA-USP14 and not another DUB contaminating the IPs that is being measured, it is essential to perform a control experiment in which a catalytically inactive USP14 mutant is used instead of the wild type protein.*

To address this question, catalytically inactive USP14 mutant was used to validate Ub-AMC assay on imminoprecipitated HA-USP14 (Figure 3—figure supplement 2). At the same protein concentration, USP14-C114S showed significantly lower catalytic activity than that of WT.

*5) In Figure 3, the effects of MK2206 on USP14 activity seem modest (compared with the effects on USP14 phosphorylation presented elsewhere).*

The extents of USP14 activity in this in vitro Ub-AMC assay stimulated by Myr-Akt (Figure 3), inhibited by MK2206 (Figure 3), or stimulated by IGF1 (Figure 3) are similar, which may be due to the limitation of this assay.

*6) In Figure 3, the S432A mutant is still activated by AKT, suggesting that Ser143 phosphorylation may cause partial activation. An experiment with Ser143Ala and Ser143Glu mutants would be informative.*

To address this question, we analyzed the activity of S143A with active Akt (Figure 3—figure supplement 2) and S143E alone (Figure 3—figure supplement 2). Our new data indicate that S143A can be activated as that of WT and S143E has no significant effect on Ub-AMC hydrolysis activity.

*7) Can USP14 activation be observed by stimulating with IGF1 in the absence of any overexpression of AKT, and if so, is it suppressed by MK2206 – this is a more critical experiment.*

To address this question, we analyzed the activity of USP14 from HEK293T cells treated with IGF-1 in the presence or absence of MK2206 and show that USP14 activity is higher when it is isolated from IGF-1 stimulated cells and the activity is inhibitable by MK2206 (Figure 3).

*8) Figure 5: Does IGF1 stimulation without AKT overexpression activate the UPS and, if so, is this blocked by MK2206?*

To address this question, we performed new experiments to show that IGF-1 stimulation (Figure 5) as well as EGF (Figure 5) could affect the UPS, which was blocked by MK2206.